# Coupling Relationship of Geomorphic Evolution and Marine Hydrodynamics in the Stage-Specific Development of Urban Bays: A Modelling Case Study in Quanzhou Bay (1954–2017), China

Xianbiao Xiao [1,2,3], Yunhai Li [1,2], Junjian Tang [1], Fusheng Luo [1], Fangfang Shu [1], Liang Wang [1], Jia He [1], Xiaochun Zou [1], Wenqi Chi [1], Yuting Lin [1] and Binxin Zheng [1,*]

[1] Laboratory for Ocean & Coast Geology, Third Institute of Oceanography, Ministry of Natural Resources, Xiamen 361005, China
[2] Laboratory for Marine Geology, Qingdao National Laboratory for Marine Science and Technology, Qingdao 266237, China
[3] Electric Power Research Institute of State Grid Fujian Electric Power Co., Ltd., Fuzhou 350007, China
[*] Correspondence: zhengbinxin@tio.org.cn

**Abstract:** With the development of social economy and human activities, the geomorphology and hydrodynamic conditions of coasts have been dramatically changed, causing serious environmental pollution and resource depletion. Taking Quanzhou Bay as an example, this study combined geomorphologic change with a hydrodynamic model to simulate the change in tidal currents in different periods. The results show a change in the coastline was the main cause of hydrodynamic change during the industrialization reform. During the past 70 years, the tidal prism decreased year by year, and the average velocity of the tidal current in the channel decreased by 33.7% and 30.8% at flood and ebb tide, respectively. In the early stages of industrialization, reclamation land was used in a single way. The tidal prism decreased by 22.2% and 29.8% in the spring and neap tide, respectively. In the middle and later stages, the tidal current velocity increased, and reclamation land was used in a variety of ways. In modern society, the reclamation land-use type was unitary. Based on this research, we show the influence of human activities on the evolution of the bay's geomorphology and provide suggestions for the management of the bay.

**Keywords:** human activities; geomorphology; coupling relationship; hydrodynamics; bays urbanization

## 1. Introduction

As a link between the sea and the land with abundant natural resources, the advantageous geographical position makes a bay play an important role in the national economic development and international cultural exchange [1]. With the increase in human activities in the past century, the main factors controlling the environmental changes in the bay have gradually changed from natural factors to human factors [2]. As the earth entered the Anthropocene, the geomorphologic evolution process of the bay has also undergone changes different from the natural conditions. Therefore, the study of the impact of human activities on geomorphologic evolution is beneficial to the sustainable development of the marine environment in the urban bay [3].

With the development of the economy, urbanization in the bay has changed the topography and hydrodynamic environment [4]. Continual human activities lead to the gradual seaward advance of the bay coastline, with decreasing bay area [5]. Previous research showed that the area of bays around the East China Sea has shrunk by 11.23% over the past 20 years [6]. Similarly, between 1987 and 2017, the areas of San Francisco Bay, the New York Bay, the Tokyo Bay, and the Guangdong–Hong Kong–Macao Bay were reduced to varying degrees [7]. These human activities (such as reclamation, channel dredging, coastal

aquaculture, etc.) not only changed the topography, but also affected the evolution of the hydrodynamic environment in the bay [8]. According to the present study of semi-enclosed bays, reclamation weakens the hydrodynamic forces and results in prolonged stays of the water at the same location [9,10]. Coastal aquaculture and port engineering have changed the local tidal flow and water structure in the bay and estuary areas [11]. This has led to increased levels of pollutants in the bay, which destroyed natural wetlands, leading to ecological deterioration around the bay [12]. Coastal aquaculture and port engineering have changed the tidal currents and water structure [11]. This has led to increased levels of pollutants in the bay, leading to ecological deterioration around the bay [12].

The hydrodynamic conditions at the estuary of the bay are closely related to the sedimentary environment of the bay, which would change the transport mode of the terrestrial materials and influence the evolution of the topography in the bay [13]. In a tidal-controlled bay with large terrestrial material flux, the sediment moves back and forth with the flood and ebb tide, and fine-grained sediments are prone to resuspension and then flocculation and deposition [14]. The reduction of most of the tidal prisms in the bay alters the hydrodynamic conditions of the bay, which, in turn, causes the movement of suspended particles in the water [10]. In the future, the transformation of the surrounding environment of the bay will be increasingly complex, and the contradiction between economic development and environmental evolution will be further highlighted [15]. Therefore, it is necessary to study the relationship between social development and the evolution of a bay's geomorphology.

At present, research regarding hydrodynamics mainly focuses on the hydrodynamic changes over short time scales and the hydrodynamic prediction after construction [7,10,11]. However, there is still a lack of research on the hydrodynamic change over the long time scale and the mechanism of sedimentary dynamics in the process of geomorphology evolution in a bay. In the past 70 years, Quanzhou has undergone the whole process of industrialization, with many periods of reclamation, resulting in significant changes in the topography of the bay [16]. There is a certain coupling relationship between the changes in the geomorphology and hydrodynamics and the development of urbanization in a bay, and this coupling relationship is obviously manifested in Quanzhou Bay, showing stage-specific development of urban bays. Therefore, Quanzhou Bay is a natural laboratory to investigate the hydrodynamic change process of a bay under the influence of human activities, and can be used as a typical area to study the relationship between the industrialization development and hydrodynamic evolution of a bay.

In this study, the coupling relationship between the geomorphologic environment of an urban bay and the hydrodynamics under the influence of human activities were analyzed by hydrodynamic model simulation. The influence of the construction of the reclamation expansion of the urban bay on the hydrodynamic and sedimentary dynamics of the bay was explored.

## 2. Materials and Methods

### 2.1. Basic Data

2.1.1. Water Depth and Topographic Data

In this study, six large-scale marine maps (1954–2017) of Quanzhou Bay were selected as the data source of the bathymetry and coastline (Figure 1, Table 1); the lowest theoretical datum was used for bathymetric data [17]. All marine map data were converted to the WGS84 coordinate system and Mercator projection (Figure 2, Table 1). Due to the dredging in Quanzhou Bay, there were some differences between the actual channel topography and the topographic data of the marine map. The accuracy and integrality of the bay's geomorphological data can be ensured by adding the actual channel data. The survey time of the channel maintenance bathymetric data was 2013, 2015, and 2017, and the main survey areas were Houzhu, Dazhui, and Neigang channels (Table 2). The survey depth datum adopted the local theoretical lowest tide level (Huanghai Vertical Datum 1956).

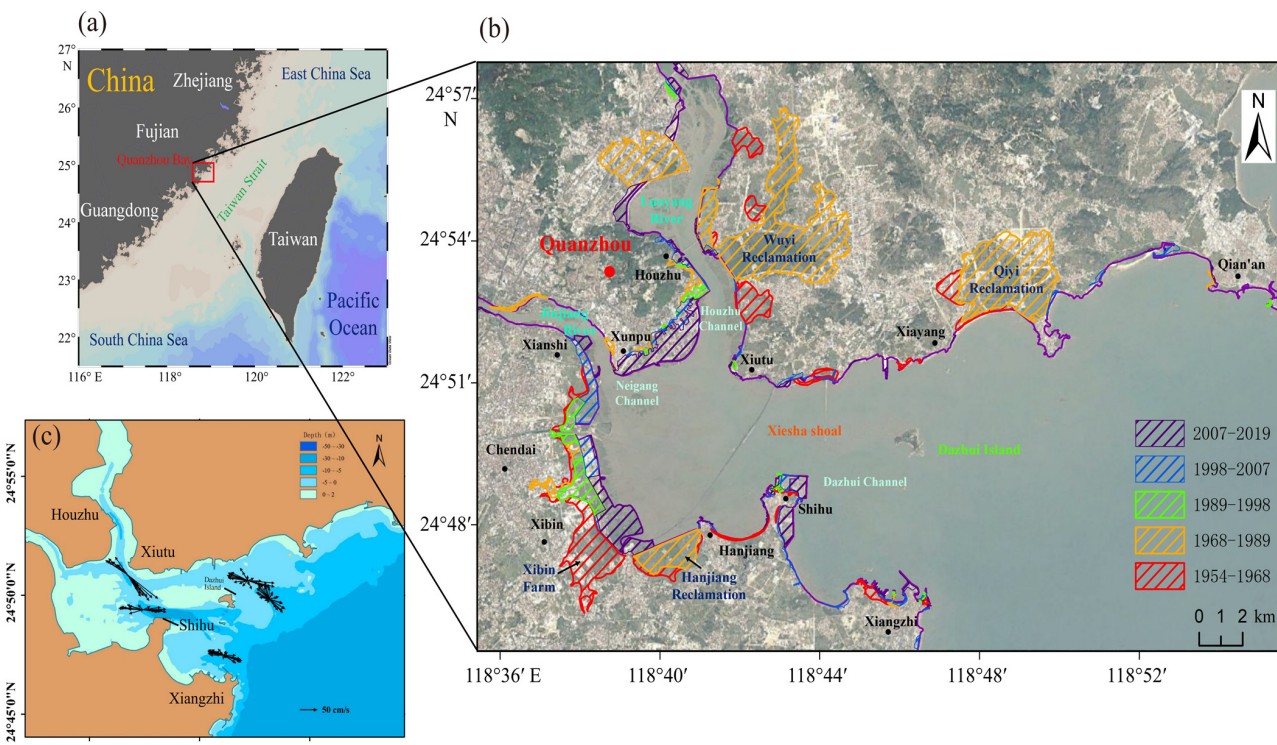

**Figure 1.** Overview of the study area. (**a**) The location of the study area. (**b**) The basic scenario of the study area, including cities, islands, channels, rivers, and reclamation areas at different stages. (**c**) Tidal currents during the Quanzhou Bay tidal cycle [18].

**Table 1.** Status of the historical marine map data.

| Marine Map Name | Marine Map Number | Scale | Coordinate System | Time of Publication |
|---|---|---|---|---|
| Nanri Island to Quanzhou Bay | 10–56 | 1:100,000 | - | April 1973 |
| Quanzhou Bay | 5616 | 1:50,000 | Beijing54 | September 1975 |
| Quanzhou Bay | 14,181 | 1:35,000 | Beijing54 | April 2000 |
| Quanzhou Bay | 14,181 | 1:35,000 | Beijing54 | April 2003 |
| Quanzhou Bay | 14,181 | 1:35,000 | WGS84 | May 2008 |
| Quanzhou Bay | 14,181 | 1:35,000 | CGCS2000 | July 2019 |

The data comes from previous studies [16].

**Table 2.** Channel maintenance time and area information.

| Measure Time | Measuring Area | | |
|---|---|---|---|
| | **Neigang Channel** | **Dazhui Channel** | **Houzhu Channel** |
| 2013 | Up to the junction of Neigang channel and Houzhu channel, down to the downstream of Quanzhou Bridge. The measuring area is about 4.2 km². | Outside to the southern part of the Dazhui island, inside to the junction of the Shihu channel and the Dazhui island channel. The measuring area is about 3.59 km². | From the Dazhui island to the Houzhu Port. |
| 2015 | Up to the Quanzhou Bridge and down to the junction of Neigang channel and Houzhu channel. | From the anchorage outside Quanzhou Bay, follow the Dazhui island channel to the ports of Shihu and Xiutu. The measuring length was 11.04 km. | It is about 8.8 km from Shihu to Houzhu. |
| 2017 | / | From the anchorage outside Quanzhou Bay, follow the Dazhui island channel to the ports of Shihu and Xiutu. The measuring length was 11.04 km. | It is about 8.8 km from Shihu to Houzhu. |

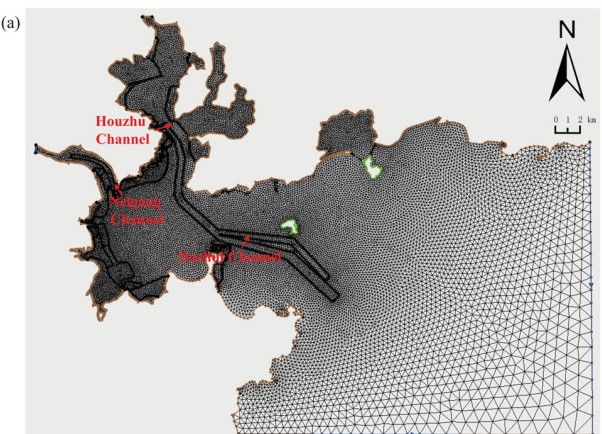
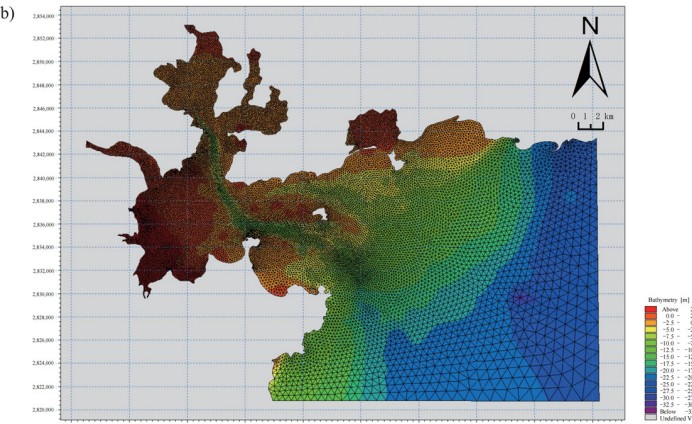

**Figure 2.** Mesh generation of the model. (**a**) Grid division of study area; (**b**) Bathymetric topography of the study area in 1954.

The shoals of Quanzhou Bay are mainly distributed from Xiutu to the west of Shihu, outside the Jinjiang River estuary, and on both sides of Luoyang River estuary. The tidal area is big, and the tidal channel is tortuous. The beaches are mainly distributed along the coasts on both sides of the bay mouth east of the Xiutu–Shihu line, with a gradient of 3°–5°. The main sandbanks are located on the west side of Dashui Island and outside the Jinjiang estuary. The underwater deep troughs are mainly distributed in the water channel from Dazhui Island to the Xiutu section [18].

In ArcGIS (Geographic Information System Information editing software developed by the Environmental System Research Institute) software, the marine map data and channel data were unified into Mercator Center projection coordinates, and the datum level was unified to the global sea level (Figure 2).

### 2.1.2. Hydrological Data

In order to ensure the accuracy of the model simulation, the engineering data were taken as the reference for the calibration of the model. The tidal level data were obtained from Chongwu and Jinyu tide stations in the Quanzhou Bay from 10:00 a.m. on 20 January 2016 to 12:00 a.m. on 9 February 2016. Tidal current velocity and flow direction data collected from five hydrological and sediment stations in the inner bay, which were observed from 10:00 a.m. on 26 January 2016 to 12:00 a.m. on 27 January 2016 (Spring tide) and from 10:00 a.m. on 3 February 2016 to 12:00 a.m. on 4 February 2016 (Neap tide). The interval of the data acquisition was 1 h. The geographical coordinates and location distribution of the tide station and hydrologic and sediment station are shown in Figure 3.

The measured tidal current data were collected by Acoustic Doppler Current Profiler (ADCP), which need to be converted into vertical mean velocity by weighted average processing. The data processing method is as follows [19]:

$$v = 0.1v_1 + 0.2v_2 + 0.2v_3 + 0.2v_4 + 0.2v_5 + 0.1v_6 \tag{1}$$

where $v$ is the average velocity of vertical line (m/s) and $v_i$ is the velocity of the different horizon stations (m/s).

### 2.2. Hydrodynamic Model of Quanzhou Bay

### 2.2.1. Model Meshing and Parameter Setting

Because the coastline of Quanzhou Bay was complex and changeable, the 1959 coastline was used as the base boundary to divide the grids in order to ensure the consistency of the grids in the calculation. The areas of the Houzhu, Neigang, and Dazhui channels was grid-encrypted (Figure 2). The coordinate system of the model adopted the Central Mercator projection. The extracted bathymetric data were interpolated into the grid in

Surface Water Modeling System (SMS), while uniformly converted into MIKE's software grid file (Figure 2).

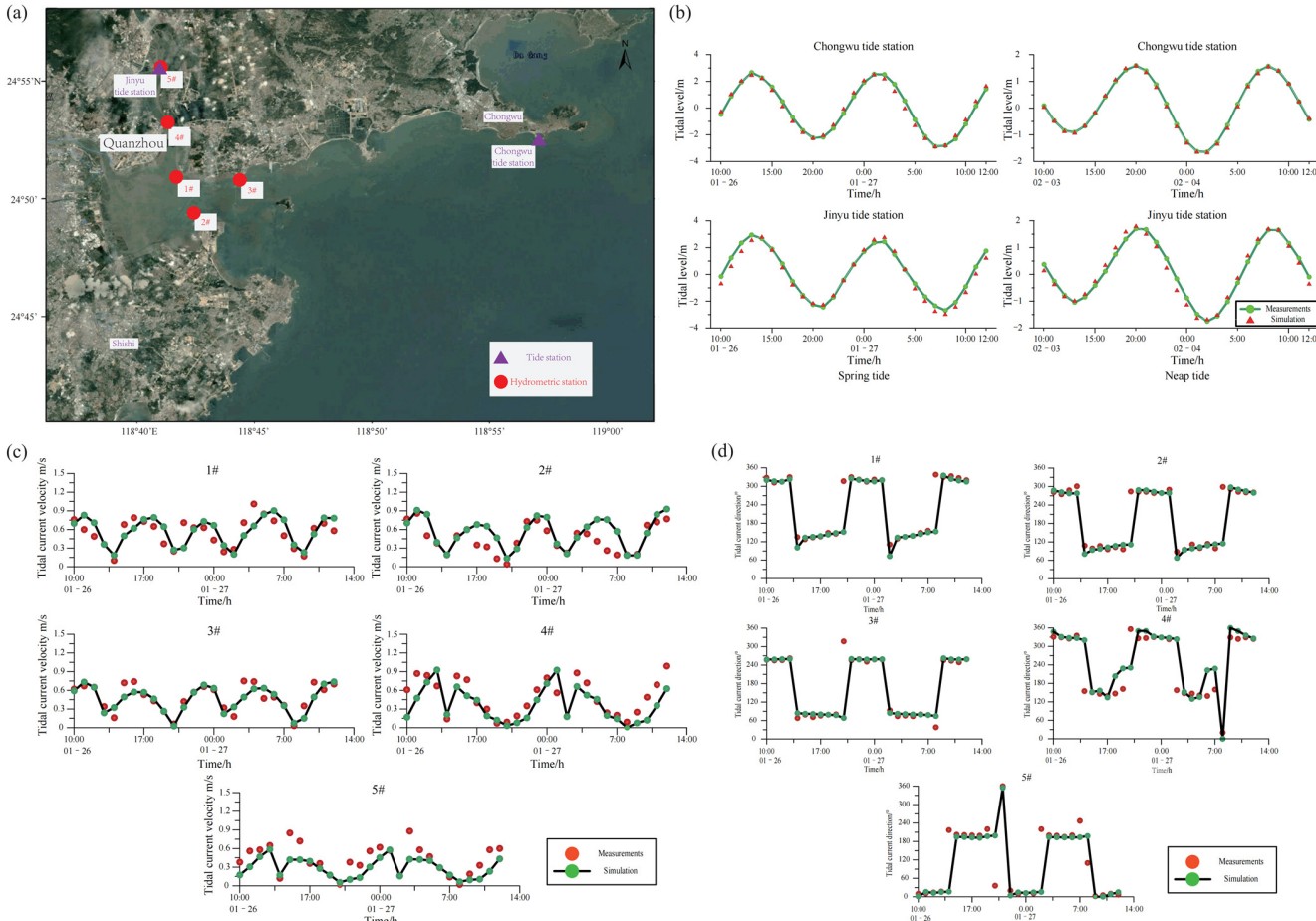

**Figure 3.** Calibration of the model. (**a**) Location of the tide stations and hydrological stations. (**b**) Comparison of the observed and model-simulated tide levels. (**c**) Comparison of the observed and model-simulated tidal current. (**d**) Comparison of the observed and model-simulated tidal current directions.

The model was simulated in a "Cold start" mode [20], with the initial water level and velocity set to 0. In this model, the Courant Friedrichs Lewy (CFL) number of the shallow water equation was set to 0.8, the main time step was 60 s, and the minimum time interval was 0.01 s. Flooding depth was set to 0.05 m, wetting depth was set to 0.1 m, and drying depth boundary was set to 0.005 m. The horizontal eddy viscosity coefficient was set to 0.28.

2.2.2. Water Exchange Capacity of the Bay

The tidal prism of the bay represents the total volume of the bay that can hold the tidal water, and can reflect the self-purification capacity of a bay [9]. The tidal prism is calculated by integrating the cross-sectional area and the product of the mean velocity of the cross-sectional area [21].

$$Q = \int_0^{t_c + \frac{T}{2}} \overline{v_i} S \, dt \tag{2}$$

where $Q$ is tidal prism, $\overline{v_i}$ is cross-sectional average velocity (m/s), and $S$ is the cross-sectional area (m$^2$).

### 2.3. Changes in Land-Use Structure

The landscape index comparison method was adopted to analyze the change in land-use structure [22]. Landscape indices (information entropy of land use structure) refers to the diversity of landscape elements or ecosystems regarding their structure, function, and change with time, which reflects the richness and complexity of green space landscape types.

$$H = -\sum_{i=1}^{n}(Pi \times lnPi) \tag{3}$$

where *H* is the landscape index, *P* is the proportion of landscape type *i* to the total land area, and *n* is the number of landscape types. In theory, when the landscape index is 0, the landscape is composed of a single element and the landscape is homogeneous; when the landscape index is the highest, the landscape is composed of more than two elements, and the proportion of each landscape type is equal. When the landscape index decreases, the difference in the proportion of each landscape type to the total land area increases.

## 3. Results

A large number of human activities have changed the coastline and underwater topography of Quanzhou Bay, which significantly affected the sedimentary dynamic environment.

### 3.1. Validation of Model Accuracy

The error was calculated (Table 3) according to the JTS-T231-2021 code for the simulation experiment of water transport engineering [23]. The model experimental deviation of tide level, velocity, and direction must meet the following requirements:

**Table 3.** Model error calculation.

| Calculation of Tidal Level Error | | | | | |
|---|---|---|---|---|---|
| **Time** | **Chongwu Tide Station** | **Jinyu Tide Station** | **Time** | **Chongwu Tide Station** | **Jinyu Tide Station** |
| 2016/1/26 13:00 | 0.08 | 0.29 | 2016/2/3 13:00 | 0.01 | 0.08 |
| 2016/1/26 19:00 | 0.09 | 0.02 | 2016/2/3 19:00 | 0.01 | 0.11 |
| 2016/1/27 1:00 | 0.09 | 0.17 | 2016/2/4 2:00 | 0.01 | 0.09 |
| 2016/1/27 8:00 | 0.18 | 0.11 | 2016/2/4 8:00 | 0 | 0.05 |
| **Calculation of Tidal Current Error** | | | | | |
| | **Time** | **1#** | **2#** | **3#** | **4#** | **5#** |
| | Flood | 0.09 | 0.01 | 0.02 | 0.17 | 0.16 |
| | Ebb | 0.09 | 0.10 | 0.03 | 0.04 | 0.01 |
| | Flood | 0.06 | 0.01 | 0.08 | 0.05 | 0.19 |
| Spring tide Mean tidal current velocity | Ebb | 0.19 | 0.22 | 0.03 | 0.08 | 0.12 |
| | Flood | 0.00 | 0.01 | 0.03 | 0.12 | 0.14 |
| | Ebb | 0.13 | 0.08 | 0.03 | 0.09 | 0.00 |
| | Flood | 0.03 | 0.04 | 0.01 | 0.11 | 0.23 |
| | Ebb | 0.17 | 0.18 | 0.07 | 0.10 | 0.10 |
| | Flood | 0.02 | 0.03 | 0.03 | 0.27 | 0.16 |
| **Calculation of Tidal Current Direction Error** | | | | | |
| | **Time** | **1#** | **2#** | **3#** | **4#** | **5#** |
| | 2016/1/26 10:00 | 7.7 | 5.5 | 0.9 | 16.6 | 8.6 |
| | 2016/1/26 11:00 | 4.7 | 6.9 | 3.4 | 2.1 | 4.9 |
| Spring tide | 2016/1/26 12:00 | 0.1 | 10.3 | 3.4 | 2.6 | 1.5 |
| | 2016/1/26 13:00 | 7.0 | 22.6 | 2.4 | 8.2 | 1.6 |
| | 2016/1/26 14:00 | 34.3 | 27.3 | 16.0 | 165.2 | 200.3 |
| | 2016/1/26 15:00 | 5.6 | 5.2 | 3.0 | 0.9 | 7.1 |

**Table 3.** *Cont.*

| | | | | | |
|---|---|---|---|---|---|
| 2016/1/26 16:00 | 2.0 | 9.4 | 10.2 | 10.8 | 6.8 |
| 2016/1/26 17:00 | 1.6 | 6.6 | 4.0 | 4.7 | 7.2 |
| 2016/1/26 18:00 | 5.9 | 4.5 | 0.1 | 56.5 | 7.4 |
| 2016/1/26 19:00 | 2.6 | 15.1 | 2.6 | 66.7 | 23.1 |
| 2016/1/26 20:00 | 165.2 | 172.6 | 248.8 | 125.2 | 163.0 |
| 2016/1/26 21:00 | 5.0 | 5.8 | 3.9 | 24.1 | 5.3 |
| 2016/1/26 22:00 | 1.3 | 5.1 | 1.1 | 23.0 | 16.0 |
| 2016/1/26 23:00 | 2.7 | 4.4 | 7.1 | 2.1 | 5.4 |
| 2016/1/27 0:00 | 6.4 | 0.1 | 0.8 | 0.1 | 0.2 |
| 2016/1/27 1:00 | 0.4 | 10.9 | 1.0 | 4.6 | 3.5 |
| 2016/1/27 2:00 | 38.2 | 20.3 | 8.6 | 165.4 | 204.3 |
| 2016/1/27 3:00 | 3.6 | 0.9 | 8.0 | 4.1 | 5.7 |
| 2016/1/27 4:00 | 0.2 | 13.5 | 4.9 | 17.3 | 5.6 |
| 2016/1/27 5:00 | 0.3 | 3.6 | 5.8 | 5.5 | 2.0 |
| 2016/1/27 6:00 | 4.5 | 5.5 | 1.0 | 83.7 | 7.2 |
| 2016/1/27 7:00 | 5.5 | 13.9 | 0.0 | 68.1 | 53.2 |
| 2016/1/27 8:00 | 184.8 | 184.2 | 36.4 | 20.0 | 87.8 |
| 2016/1/27 9:00 | 5.4 | 6.0 | 6.2 | 31.0 | 2.1 |
| 2016/1/27 10:00 | 9.3 | 7.5 | 4.6 | 25.7 | 5.4 |
| 2016/1/27 11:00 | 8.3 | 4.2 | 8.9 | 6.7 | 1.9 |
| 2016/1/27 12:00 | 4.8 | 1.3 | 0.2 | 1.7 | 8.0 |

Tide level: the deviation of high and low tide time and phase must be within $\pm0.5$ h; the maximum and minimum tide level deviation must be less than $\pm0.1$ m.

Velocity and direction of tide: the deviation of the mean velocity must be within $\pm10\%$; the shape of the velocity process line must be basically consistent; the deviation from the main tidal current direction must be within $\pm10°$.

The simulation results (Table 3) show that the calculated data of the Chongwu and Jinyu Tide Stations were basically consistent with the measured data. The individual error exceeding $\pm0.1$ m was caused by the small water depth of the Jinyu station located in the inner bay. The error between the measured value and the simulated value at other time is less than 0.1 m, which indicates that the model has a good fitting degree. The error between the measured and simulated tidal levels at high and low tides was less than 0.1 m, indicating that the model was relatively accurate for tidal level simulation. Station 5# is located in the northern side of the inner bay with a shallow water depth, and its depth varies greatly in the ebb and flood tide. This results in a discrepancy between the observed and simulated values at the site in the simulation of the speed and direction of tidal flow. The velocity process line basically coincides with the simulated value; the velocity and direction curves of the other 4 stations were basically consistent with the measured ones. The error in tidal current direction was kept within $\pm10°$, and part of the larger error was mainly caused by the delay in the instrument measurement when the tidal currents was reversed. According to the test results, the simulation results of the model were accurate.

*3.2. Variation of Exchange Intensity of Bay Water*

Based on the hydrodynamic model, the tidal prism of Quanzhou Bay under different topographic conditions was calculated. The calculation area included the line from Xiangzhi to Qian'an to the Luoyang River and Jinjiang River in the bay (Figure 4). The calculated results show that the tidal prism decreased year by year from 1954 to 2017, and the total tidal prism decreases were $2.902 \times 10^8$ m$^3$ and $2.592 \times 10^8$ m$^3$ during the spring and neap tides, respectively. The change in tidal prism from 1968 to 1988 was the largest, and during this period, the tidal capacity decreased by $1.554 \times 10^8$ m$^3$ and $1.401 \times 10^8$ m$^3$, accounting for 53.5% and 54.1% of the total variation, respectively.

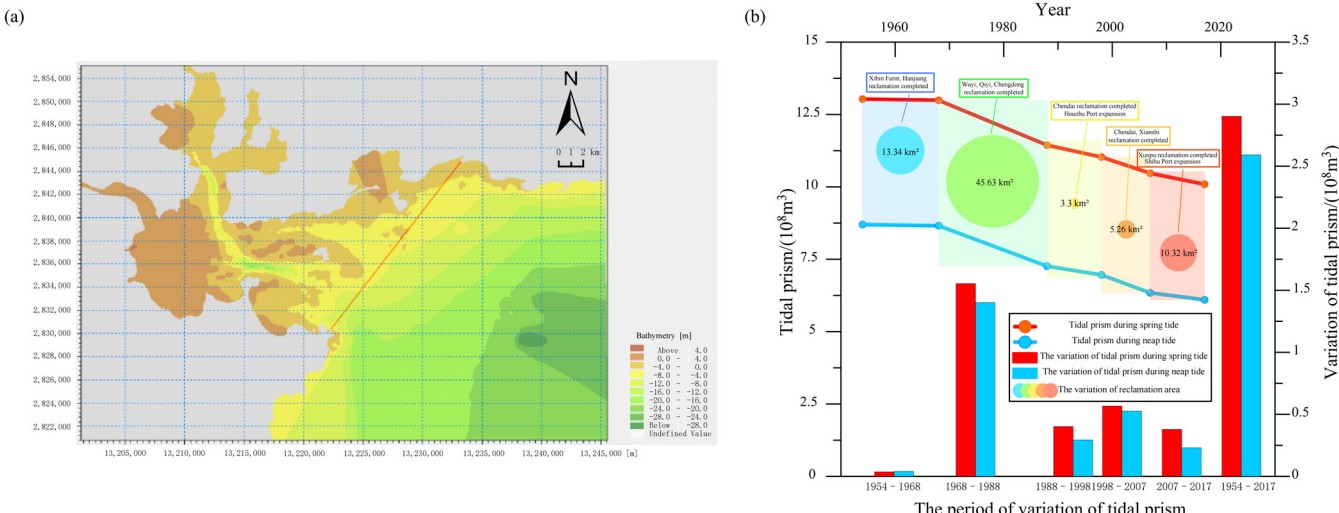

**Figure 4.** Tidal prism in different periods. (**a**) The calculation range of the tidal prism. (**b**) The tidal prism of Quanzhou Bay in different periods and reclamation at the same time.

### 3.3. Variation of Tidal Current Field

3.3.1. Change of Flow Field in Different Period

A model was used to simulate the characteristics of the tidal current field in Quanzhou Bay at different time periods (Figure 5). At flood tide time, the tidal current entered the inner bay from the outer bay along the North and South channels around Dazhui Island, and converged on the south side of Xiutu and then reached the Luoyang River and the Neigang channel along the Houzhu channel. In 1954, the maximum tidal current velocity appeared in the west of Xiutu and the north of Shihu. At that time, the tidal current velocities along the Xibin were all below 0.15 m/s, and the tidal current velocity in the tidal flat area of the lower Jinjiang River was between 0.15 and 0.75 m/s. Until 1968, the maximum tidal current velocity during flood tide still appeared in the Houzhu–Dazhui channel area, but the tidal current velocity decreased. The current velocity in the tidal flat region of the lower Jinjiang River was about 0.9–1.2 m/s. From 1968 to 1998, the current velocity in the channel area decreased, but the direction of the coastal tidal current changed from SSW to WNW. By 2007, the Neigang channel was opened for navigation, at which time the maximum tidal current velocity of the flood tide increased from 1.5 to 1.8 m/s. In 2017, the maximum tidal current velocity during flood tide still occurred in the northern side of Shihu. The same flow field changes during the ebb tide are shown in Figure 5.

3.3.2. Variation of Current Velocity in Channel Area

According to the variation in tidal current velocity and the position during flood tide, the area with a greater hydrodynamic difference was mainly from the Houzhu to Dazhui channels. The hydrodynamic changes in the outer bay area were small and stable. In order to estimate the hydrodynamic change in the channel area, the average current velocity in across channel's characteristic points during the flood and ebb tide was calculated by using the data of the point velocity in the channel area simulated by the model for different periods (Figure 6).

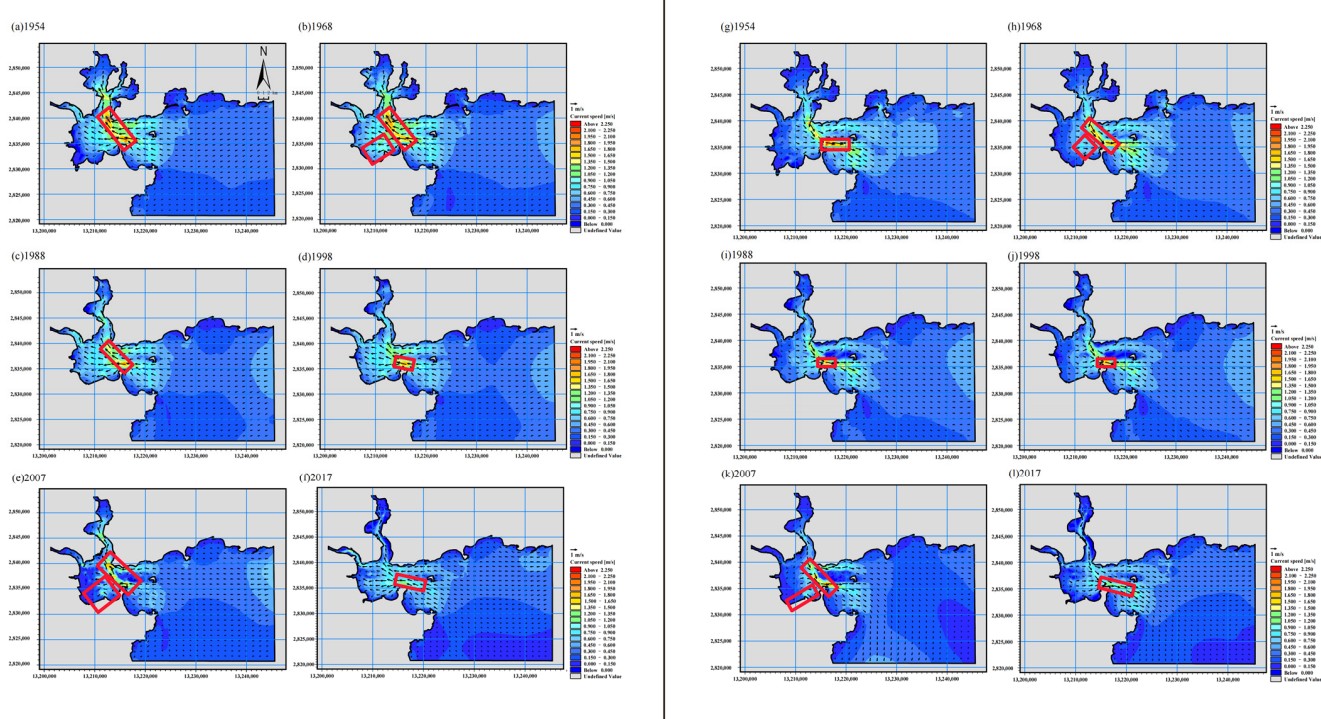

**Figure 5.** Changes in the tidal current field of the rising and falling tides in Quanzhou Bay during different periods. (**a–f**) The distribution of the hydrodynamic forces during the flood tide in different years. (**g–l**) The distribution of hydrodynamic forces during the ebb tide in different years.

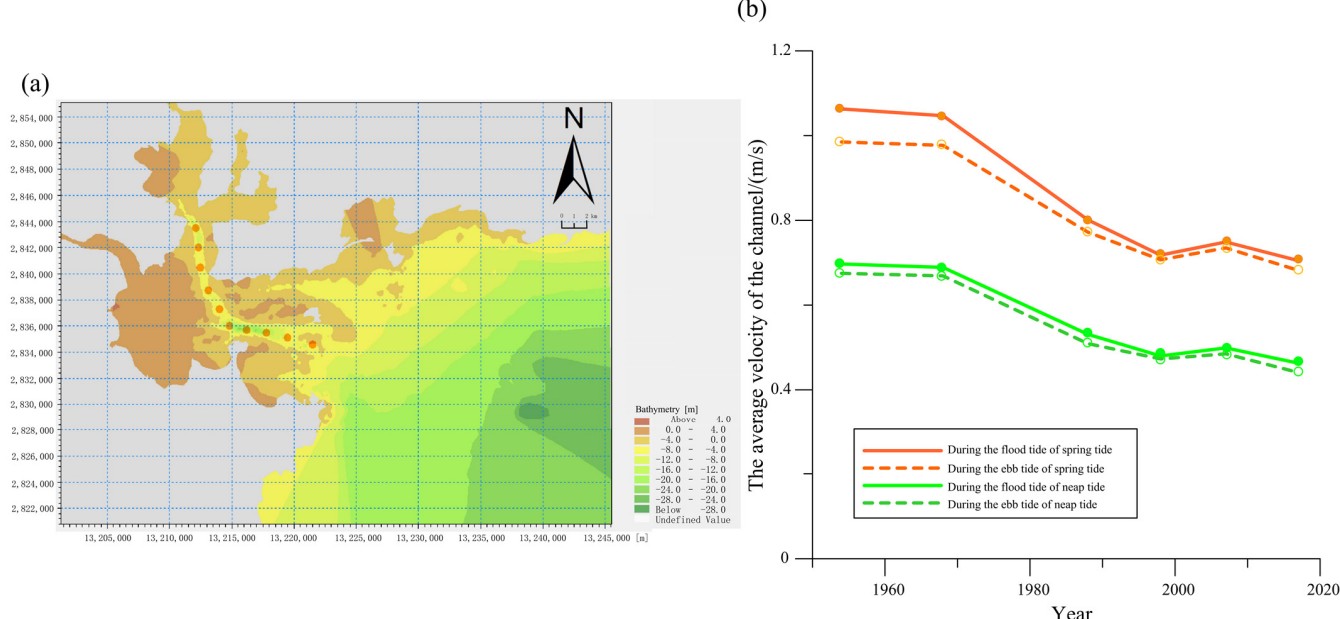

**Figure 6.** Changes in tidal current field of the flood and ebb tides in Quanzhou Bay during different periods. (**a**) Tidal current velocity calculation point distribution location; (**b**) The average velocity of the channel area in different years.

The results show that the average tidal current velocity in the channel area decreased continuously from 1954 to 1998 (Figure 6). During the spring tide from 1968 to 1988, the large-scale reclamation projects resulted in the hydrodynamic weakening of the inner bay. At this time, the average velocity during the flood and ebb tide in the channel decreases

from 1.047 m/s and 0.977 m/s to 0.801 m/s and 0.772 m/s, decreasing by 23% and 21%, respectively. From 1998 to 2007, the navigable channel led to an increase in water depth in the channel area, and the flow converged in the channel area. During the spring tide, the average current velocity of the channel during flood tide and ebb tide increased by 0.031 m/s and 0.028 m/s, respectively. In 2017, the current velocity decreased to 0.705 m/s and 0.682 m/s, respectively.

## 4. Discussion

### 4.1. Effects of Human Activities on the Geomorphology and Hydrodynamic Evolution of the Bay

4.1.1. Influence of Sea Area Change on Exchange Capacity of Bay Waters

According to the calculation results of the model, it can be found that the main reason for the decrease in the tidal prism was the reclamation project. Reclamation reduced the area of the bay, which reduced the flow of tidal currents into the bay (Figure 4). From 1954 to 1968, the height of the shoals around the reclamation project was higher than the mean sea level. Although the reclamation area was large (Table 4), the impact of the reclamation on the variation in tidal capacity of the bay was minimal. During 1968 to1988, large-scale reclamation projects, such as "Wuyi reclamation" and "Qiyi reclamation", were carried out at this time, which directly blocked the tide from entering north of Baiqi village. As a result, the tidal prism was greatly reduced at this time. The reduction in tidal prism accounted for 52.7% and 53.8% of the total reduction. From 2007 to 2017, dredging in the channels and returning farmland to sea were carried out, which slowed down the trend of tidal prism reduction during this period.

Large-scale reclamation often results in a decrease in the area of the intertidal zone of the bay, reducing the tidal prism, which, in turn, will weaken the self-purification capacity of the water body [24]. Nevertheless, the development of industrialization has promoted the development of printing and dyeing, textile, electroplating, and other industries around the urban bay. Part of the industrial waste water and the sewage of the cities around the bay were discharged through sewage pipes or open channels, leading to a continuing increase in organic matter and heavy metal elements (such as Zn, Pb, etc.) in the bay [25,26]. In addition, the decrease in tidal prism prolonged the retention time of heavy metals and led to the chronic accumulation of pollutants, which had worsened the water quality of the bay [27]. With the continuous deposition of the pollutants in the bay, the retention of some nutrients and other artificial compounds may cause frequent red tide outbreaks [28]. This will affect the survival and reproduction of the original marine organisms, and also have an impact on coastal aquaculture [29,30], which makes the city's bay water quality problems increasingly significant.

**Table 4.** Reclamation area and landscape indices of Quanzhou Bay at different stages.

| Year | Cities and Industries (m²) | Agriculture (m²) | Port (m²) | Aquaculture (m²) | Other Types of Land (m²) | Total Area for Reclamation (m²) | Landscape Indices |
|------|------|------|------|------|------|------|------|
| 1972 | 5.64 | 6.71 | 0 | 1.01 | 0.23 | 13.588 | 0.976 |
| 1988 | 12.6 | 39.1 | 0.80 | 1.39 | 2.74 | 56.628 | 0.888 |
| 1995 | 17.95 | 33.75 | 1.29 | 4.20 | 3.00 | 60.188 | 1.103 |
| 2006 | 27.21 | 25.82 | 2.22 | 7.06 | 4.07 | 66.378 | 1.256 |
| 2019 | 41.02 | 22.61 | 3.69 | 5.56 | 6.09 | 78.968 | 1.226 |

4.1.2. Variation of Current Velocity in Channel Area

According to the hydrodynamic simulation results of Quanzhou Bay across different periods, the hydrodynamic forces in the bay as a whole appear to have weakened. Since the coastline and topography of Quanzhou Bay in different periods are different in the marine map information, the hydrodynamic changes may be related to both the coastline and the topography. In order to explore the influence of coastline change on the hydrodynamics

of Quanzhou Bay, a hydrodynamics simulation was carried out by changing the coastline condition and keeping the water depth unchanged.

By comparing the simulated results of the changing coastline with the original hydrodynamic distribution (Figure 7), it was found that the coastline change has a significant impact on the hydrodynamics. In 1968, with the completion of the expansion of the military reclamation farms in the lower Jinjiang River and Xibin Farm, the average tidal current along Xibin decreased (Figures 5 and 7). There was a decrease in the tidal current velocity in the Dazhui channel area in northern Shihu. In 1988, after the "Wuyi reclamation" and "Qiyi reclamation" projects were completed, the tidal current of the channel from Houzhu to Dazhui decreased significantly. The tidal current velocity of Gufu and Xibin also decreased. Due to the completion of "Qiyi reclamation", the current velocity in the coastal area of Zhangban decreased. From 1998 to 2017, the excavation of the Neigang channel caused part of the water flow in Jinjiang River to converge with the water flow in the Houzhu channel on the western side of Xiutu, and the tidal currents in the northwest and western Shihu channels also decreased. Similarly, the contrast between the coastline changes and hydrodynamic distributions during neap tides is shown in Figures 5 and 7.

The variation in coastline has a greater impact on the hydrodynamics. As the width of the coast decreased, coastal currents increase and the seabed was continuously scoured, which will further deepen the passage between the coasts, such as the bay of Visant in France [31]. In some bay mouths, the reclamations have resulted in changes in the direction of the local flow fields, resulting in erosion of coastlines and seabed in the region under local wind waves and long-term natural storm surges [32]. At the same time, reasonable reconstruction of the bay line combined with the specific hydrodynamic conditions in the region can effectively prevent the erosion of the coastline [33]. Well-structured hydraulic structures (such as spur dikes and protective levees) can also ameliorate the problem of coastal deposition to some extent [34].

### 4.2. The Stage Development of Urban Bay Hydrodynamics and Geomorphology Evolution

According to previous research, the development of industrialization can be divided into the early stage, middle stage, late stage, and modern society [18]. According to the research results, we can find that the geomorphology and hydrodynamics of the bay show different characteristics at different development stages of the urban bay.

Before and during the early stage of industrialization, the geomorphologic changes of the bay were mainly controlled by natural factors. Marine hydrodynamic erosion and coastal sediment deposition were the main causes of coastline changes [35,36]. Before 1968, the tidal current velocity of Points 1#–3# in the river channel decreased due to the natural siltation of the river channel in the Houzhu Port area (Figure 8). In the Shihu Port area, there was almost no reclamation around the area, and the geomorphological changes were mainly influenced by natural factors. The regional tidal current velocity in the northern Shihu Port was high, and the channel presented a state of erosion. The tidal current velocity in Point 1# and 3# changed very little, and the tidal current velocity in Point 2# decreased due to the near-shore silting area. At this time, the reclamation land use was similar to that of most cities, which was mainly used for agricultural land (Table 4). As a result, the landscape index was low, and the reclamation land use was unitary.

From 1968 to 1998, the Quanzhou Bay was in the early to mid-stage of its industrialization. Large-scale reclamation activities such as "Wuyi reclamation" and "Qiyi reclamation" were carried out (Figure 4). The reclamation dam built on the west side of Baiqi village during the "Wuyi reclamation" project prevented water from entering the bay, resulting in a decrease in the influx of tidal water, which further weakened the originally enclosed hydrodynamic environment of Quanzhou Bay (Figure 8). Meanwhile, the construction of reclamation dams for reservoirs and sluices upstream of the Jinjiang and Luoyang Rivers has reduced the runoff and weakened the impact of runoff on tidal movements. Under the action of tidal current, some sediment particles ascend to the area of the Houzhu channel with tidal current due to re-suspension. Due to the reduced hydrodynamic forces in the

bays, tidal forces were not sufficient to transport sediment particles out of the Houzhu channel, resulting in deposition in the channel area [37]. The development of the tertiary sector of the economy had increased the shipping demand and prompted the construction of new berths at the Shihu Port. The expansion of the berth extended the shoreline of the port and changed the hydrodynamic direction (Figure 8). Deposition of the channel reduced the tidal hydrodynamic forces and the narrow pipe effect of the tidal channel between the inner and outer bays [38,39], which attenuates ocean erosion around the Xiesha shoal. Due to the weakening of the hydrodynamic forces in the inner bay, the bay mouth was mainly eroded by ocean hydrodynamic forces, resulting in the landward movement of the 20 m isobath (Figure 8). At this time, the reclamation land increased, and the purpose of land use became more obvious with the land mainly used for agriculture, and the landscape index dropped (Table 4).

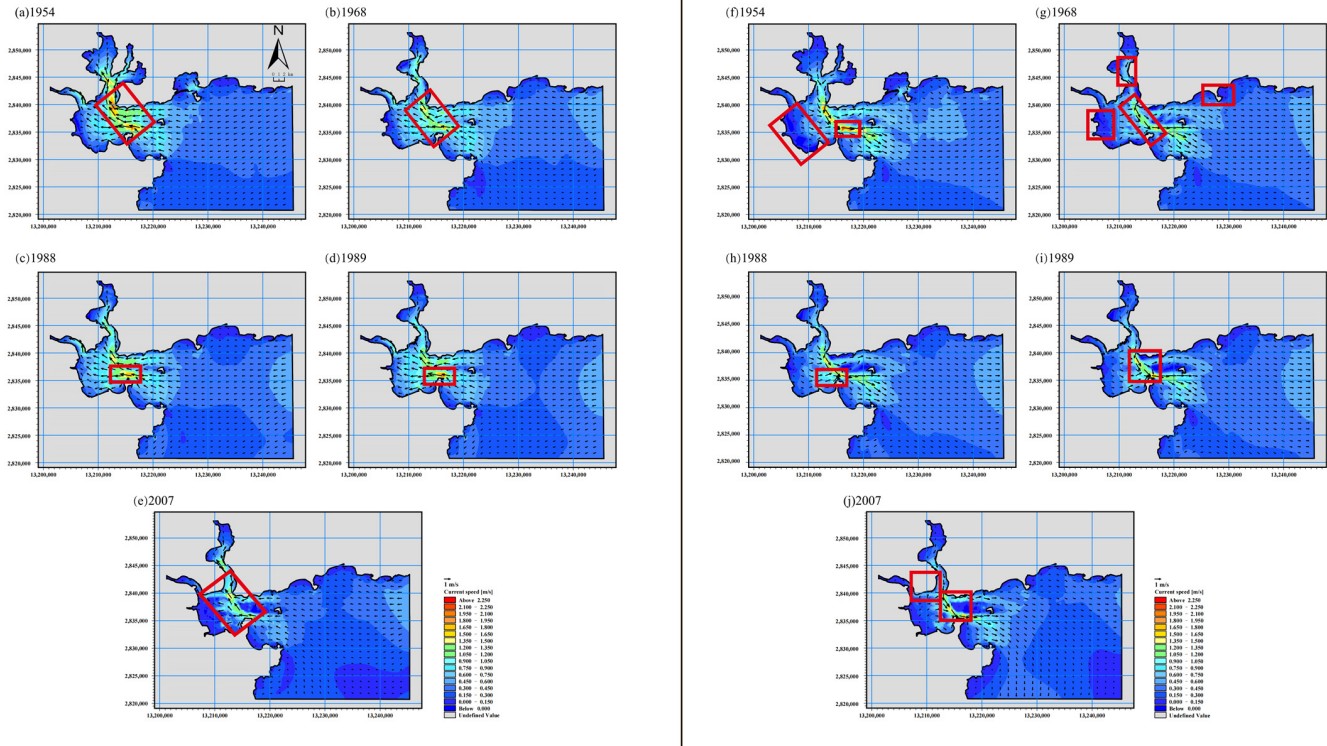

**Figure 7.** Hydrodynamics changes of Quanzhou Bay caused by coastline change over different periods. (**a–e**) The distribution of hydrodynamic forces during the flood tide in different years after the changes in the coastline. (**f–j**) The distribution of hydrodynamic forces during the flood tide over different years after the changes in the coastline.

From 1998 to 2007, the Quanzhou Bay was in the late stage of industrialization. The tidal current velocity decreased slightly in Points 1# and 2# of the Houzhu port, whereas that in Point 3# remained unchanged. The channel deposition thickness continued to rise. The excavation of the Neigang channel increased the influence of runoff on the tidal current, which increased the tidal current velocity. With the economic development and industrial upgrading, the area of industrial land, port land, and aquaculture land in the reclaimed land increased, which made the landscape index rise, with the types of reclaimed land showing the characteristics of diversification (Table 4).

During the period of 2007–2017, several channel dredging projects resulted in the re-deepening of the water depth, which caused the flow velocity in Houzhu channel to change again. This weak hydrodynamic environment increased the risk of deposition in the channel and affected the movement of ships in the channel. The Shihu ports continued to expand eastward. Under the action of the tides, a slow flow zone was formed in the

bay. The sediment that was originally deposited was suspended under the convolution of the coastal waves [40], which migrated with the waves to the slow flow zone of the shoal. With the decrease in the whole hydrodynamic force in the bay area, a new deposit area was formed in the area around the Xiesha shoal, which made the Xiesha shoal area expand continuously. During the period from late industrialization to modern society, the reclamation land around the bay was mainly used for ecological restoration and port expansion. At this time, the reclamation land was mainly used for ecological restoration and urban development; the single use of land made the landscape index decrease.

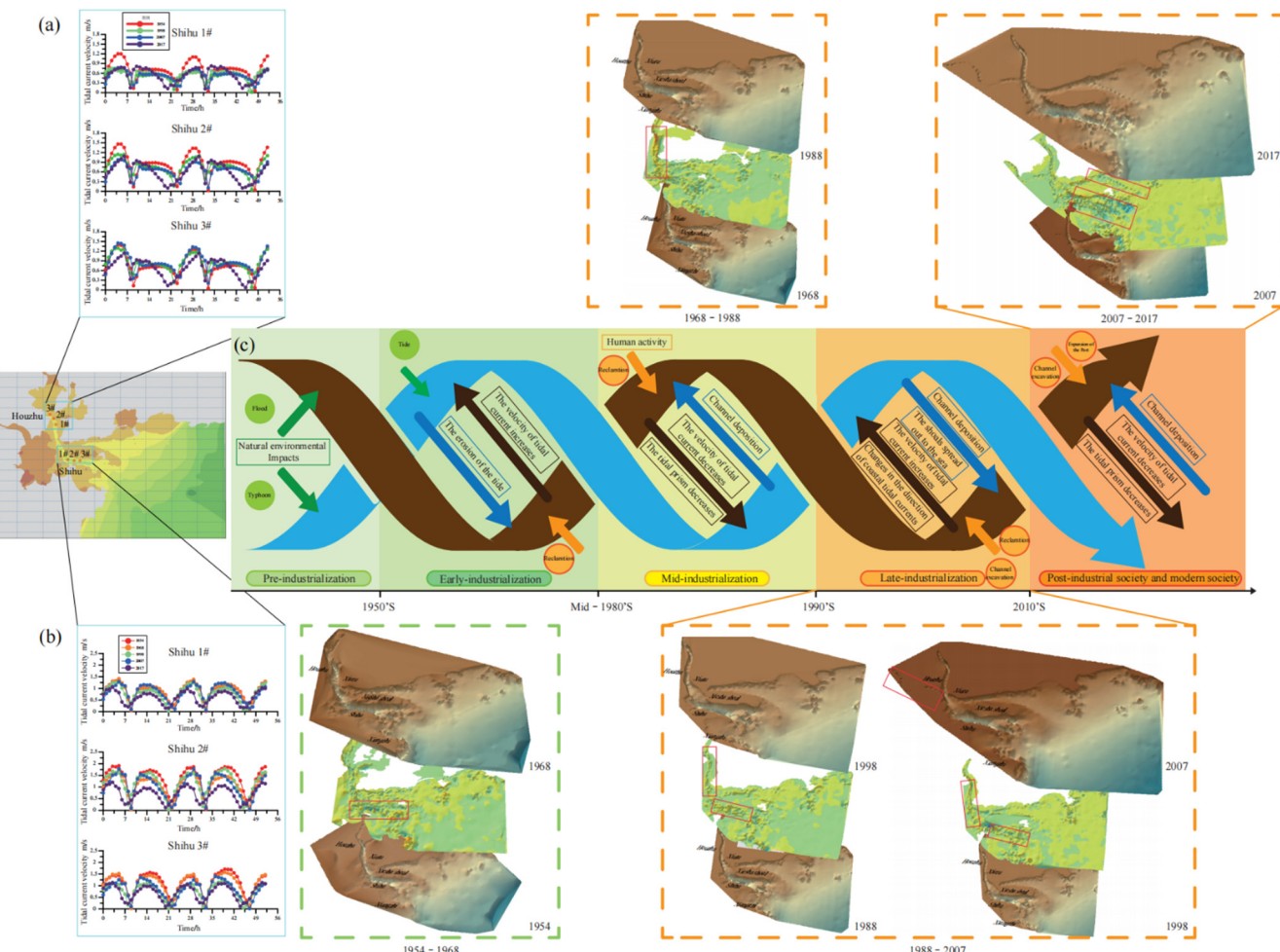

**Figure 8.** Coupled hydrodynamics and geomorphologic evolution in Quanzhou Bay. (**a**) Tidal currents in the Houzhu channel at different periods. (**b**) Tidal currents in the Shihu Port at different periods. (**c**) Coupling of geomorphologic changes and hydrodynamics in the Quanzhou Bay. The blue arrow shows the hydrodynamic development and the brown arrow shows the development and change of the bay geomorphology.

The industrialization of the bay has changed the shape of the coastline and sea area as well as changed the original conditions of the water and sediment transport, which caused changes in the hydrodynamic and sedimentary environment. The coastal construction of urban bays transforms most natural coastlines into artificial coastlines and changes the original hydrodynamic intensity and direction, which affects the natural development of topography in the bay. The coupling relationship between the sedimentary environment and hydrodynamics is a key to the future evolution of the bay geomorphology; indeed, the coupling relationship between the human activities and geomorphology has existed in other bays in the past, such as tidal flat deposition in the Pearl River estuary [41,42].

*4.3. Future Urban Bay Development and Planning*

With the continuous intervention by human activities, most urban bays have experienced different degrees of channel deposition and environmental pollution in bays. Currently, in China, the number of channels in most urban bays that do not meet the design standards is rising [43]. From the hydrodynamic model simulation results of Quanzhou Bay and the research status of other bays, it can be seen that repeated dredging projects fundamentally cannot solve the problem of channel deposition [44,45]. Therefore, more measures need to be taken to repair and transform the ecological environment of the urban bay to fundamentally solve the problem of the continuous deterioration of the bay environment.

The pollutants discharged into the bay are readily adsorbed by particles and then deposited in the bay [46,47]. Therefore, reducing the discharge of pollutants is the important way to reduce the pollution in the bay. For economically developed urban cities, large heavy industrial enterprises should upgrade and innovate industrial technologies and eliminate some backward industries, so as to reduce the input of heavy metals and artificial compounds into the bay [48]. This is of great significance to coastal agriculture and the surrounding ecological environment. For urban cities that are still in the process of industrialization, the carrying capacity of the bay environment should be considered in advance when expanding the scale of the enterprises around the bay [49].

In the area of ecological restoration, it is necessary to develop a regional development strategy to coordinate the development of the bay and regional ecological environment [1]. The coastal vegetation can absorb and transport heavy metal particles to a certain extent, so the establishment of nature reserves to restore the coastal vegetation has good ecological benefits [50]. The economically developed bay area should repair the damaged ecological environment in time. The economically underdeveloped bay area should consider adopting a balance of nature development options. The ecological development of the bay is combined with energy-saving and emission-reduction projects; examples include China's "Carbon neutrality", the "Blue Bay" restoration project and its "Peak carbon dioxide emissions" strategy, and offshore wind projects under development in many countries [51].

For the development policy, the government needs to plan development projects and activities and formulate the relevant legal policies to protect the urban bay ecosystem. Most of the economically developed bays are now working to reduce the adverse impacts of reclamation on the bays by setting up specialized bays regulatory agencies and establishing the relevant legal provisions, measures such as the establishment of a central government-led regulatory commission and the establishment of a marine coordination regulatory agency in the Netherlands, the enactment of the Public Surface Burial Act in Japan, and the Coastal Zone Management Act in the United States, to manage reclamation land [52]. In addition, the re-planning of completed reclamation is also an important tool for the restoration of the bay, as is the planning for the reconstruction of parts of old industrial zones in Spain [53]. However, it is also necessary to consider the differences between the socio-economic development levels and select the appropriate indicators to evaluate the development of the bay.

## 5. Conclusions

Based on the topographic and hydrological data of the bay, a numerical model was used to simulate the hydrodynamics of the bay under the influence of human activities. Finally, the paper discusses the coupling relationship between human activities and the evolution of geomorphology and hydrodynamics in the urban bay at different stages of industrialization. The main conclusions are as follows:

First, reclamation during the industrialization period are the main reasons for the decrease in tidal prism in the bay. During the whole industrialization period, with the increase in reclamation area, the water exchange capacity of Quanzhou Bay decreased. The tidal prism in Quanzhou Bay decreased 22.2% and 29.8% in the spring and neap tide, respectively.

Then, the main location of the hydrodynamic changes was in the channel of the bay. The mean tidal current velocity in the channel shows a trend of first decreasing, then increasing, and then decreasing.

Finally, there is an obvious coupling relationship between the geomorphologic evolution and the hydrodynamic forces in the urban bay, and the whole process of industrialization presents a special stage evolution process. There is an obvious coupling relationship between the geomorphologic evolution and the hydrodynamic forces in the urban bay, and the geomorphology shows a special stage evolution process in the whole process of industrialization. Before and during the early stage of industrialization, the geomorphologic changes of the bay were mainly influenced by natural factors. In the middle of industrialization, the coastal reclamation project increased the sediment content and weakened the hydrodynamic force in the bay, resulting in a large deposition in the channel area. The sedimentation of the seabed made the current velocity decrease, and formed a weak hydrodynamic environment in the channel area, which made the channel deposition more serious. In the late period of industrialization, the excavation and navigation of the Neigang channel made the flow of the channel converge, and the hydrodynamic force of the channel increased. In 2014, the further expansion of the reclamation area led to a decrease in hydrodynamic force. A reduced water exchange capacity allowed heavy metals and synthetic materials to deposit in the bay, causing changes in sediment composition, while human activities such as sand mining and dredging projects directly altered the underwater seabed. In future bay development, it is necessary to combine the specific situation of the bay with certain model simulation and prediction before development.

This research clarifies the stage evolution process of an urban bay in the process of industrialization, which can provide suggestions for the future development of the bay and ensure the sustainable development of an urban bays.

**Author Contributions:** Conceptualization, Y.L. (Yunhai Li) and X.X.; validation, J.T. and F.L.; formal analysis, F.S.; investigation, J.H. and L.W.; resources, B.Z.; writing—original draft preparation, X.X.; writing—review and editing, X.Z., W.C. and Y.L. (Yuting Lin). All authors have read and agreed to the published version of the manuscript.

**Funding:** This research was funded by the National Science Foundation of China (41976050, 42176220), and the Scientific Research Foundation of the Third Institute of Oceanography, MNR (TIO2019028, TIO2015014).

**Institutional Review Board Statement:** Not applicable.

**Informed Consent Statement:** Not applicable.

**Data Availability Statement:** Data inquiries can be directed to the corresponding author.

**Acknowledgments:** Many thanks to the editors and reviewers for their suggestions and amendments to this manuscript.

**Conflicts of Interest:** The authors declare no conflict of interest.

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
