# Peer review of "Coupling Relationship of Geomorphic Evolution and Marine Hydrodynamics in the Stage-Specific Development of Urban Bays: A Modelling Case Study in Quanzhou Bay (1954–2017), China"

_jmse, doi:10.3390/jmse10111677_

Round 1

Reviewer 1 Report

I have carefully read the paper and I believe that it needs a detailed revision concerning its language. The study is interesting, and it is worth publishing. This study is investigating the hydrodynamics of the Quanzhou Bay under the influence of human activities using a numerical model simulation. Furthermore, is recording the coupling relationship between human activities and the evolution of geomorphology and hydrodynamics in the Quanzhou Bay at different stages of industrialization.

General comments:

1.   The manuscript follows the MDPI template.

2.   The abstract provides sufficient information about the manuscript content.

3.   The use of the English language needs a detailed revision.

4.   The references are well documented throughout the manuscript.

5.   Most of the figures needs to be slightly upgraded

6.   Maps are proper with good quality but they are small in size.

For specific comments, please refer in the manuscript.

Overall, my suggestion is that the paper should be published after minor improvements.

Author Response

  1. The manuscript follows the MDPI template.

Answer: Thank you for your comments.

  1. The abstract provides sufficient information about the manuscript content.

Answer: Thank you for your comments.

  1. The use of the English language needs a detailed revision.

Answer: Thank you for your comments.

We have made a detailed revision of the language in the manuscript and some of the changes can be found in file named ‘Revised manuscript with marks’.

  1. The references are well documented throughout the manuscript.

Answer: Thank you for your comments.

  1. Most of the figures needs to be slightly upgraded.

Answer: Thank you for your comments.

We added north arrow and scale in the figure.

  1. Maps are proper with good quality but they are small in size.

Answer: Thank you for your comments.

We have uploaded high quality and large size figures.

For specific comments, please refer in the manuscript.

Answer: Thank you for your comments.

We checked the specific comments in the manuscript and modified the comments. Please see the file named ‘Revised manuscript with marks’ for details of the modifications.

Reviewer 2 Report

The manuscript shows an example, that combines the geomorphologic changes caused by human activities with the hydrodynamicmodel to simulate the hydrodynamic changes in different periods.

The manuscript is very interesting and well written. The final model is credible, but, from a geological point of view, I suggest to add:

1- a description of the substrate or, if the characteristics of the substrate do not affect the model, the authors should explain why.

2 - a description of the characteristics of the sediments (compositional and / or grain-size) and of any changes over time of the sedimentary filling.

Minor comments:

- correct the abstract, there are some words repeated too many times in the same sentences.

- In the text change for example: (Figure. 1) with (Fig. 1) , following the rules of the journal.

- In the captions of the figures change (a), (b) in bold, following the rules of the Journal.

Author Response

Reviewer #2:

The manuscript shows an example, that combines the geomorphologic changes caused by human activities with the hydrodynamicmodel to simulate the hydrodynamic changes in different periods.

The manuscript is very interesting and well written. The final model is credible, but, from a geological point of view, I suggest to add:

  1. A description of the substrate or, if the characteristics of the substrate do not affect the model, the authors should explain why.

  Answer: Thank you for your comments. 

  We add a description of the study area substrate in the Basic data.

  The shoals of Quanzhou Bay are mainly distributed from Xiutu to the west of Shihu, outside the Jinjiang River estuary and on both sides of Luoyang River estuary. The tidal area is big and the tidal channel is tortuous. The beaches are mainly distributed along the coasts on both sides of the bay mouth east of the Xiutu-shihu Line, with a gradient of 3°-5°. The main sandbanks are located on the west side of Dashui Island and outside the Jinjiang estuary. The underwater deep troughs are mainly distributed in the water channel from the Dazhui Island to the Xiutu section. 

  1. A description of the characteristics of the sediments (compositional and / or grain-size) and of any changes over time of the sedimentary filling.

  Answer: Thank you for your comments.

  This is a good question. Changes in sediment characteristics, such as grain size and regional distribution, have been considered.. However, sediment data from ten years ago in the study area are scarce. we have only collected data from recent years, which can not be compared systematically. Secondly, the difference between the methods used to measure sediment ten years ago and those used today makes it impossible to directly compare the results. Finally, due to the dredging of the channel many times in the past, The sediment data collected in recent years are very different in size and location distributions and can not be compared. Therefore, the changes of sediment characteristics are not discussed in this manuscript.

Minor comments:

Correct the abstract, there are some words repeated too many times in the same sentences.

  Answer: Thank you for your comments.

  We edited the abstract to remove repeated words from the sentences.

In the text change for example: (Figure. 1) with (Fig. 1), following the rules of the journal.

  Answer: Thank you for your comments.

  We changed the text of the picture according to the rules of the journal.

In the captions of the figures change (a), (b) in bold, following the rules of the Journal.

  Answer: Thank you for your comments.

  We put the figures of the pictures in bold according to the rules of the journal.
